# The Phenomenon of Thrombotic Microangiopathy in Cancer Patients

**DOI:** 10.3390/ijms25169055

**Published:** 2024-08-21

**Authors:** Alexander Vorobev, Victoria Bitsadze, Fidan Yagubova, Jamilya Khizroeva, Antonina Solopova, Maria Tretyakova, Nilufar Gashimova, Kristina Grigoreva, Sabina Einullaeva, Maria Drozhzhina, Aygun Hajiyeva, Emilia Khalilulina, Alexander Cherepanov, Daredzhan Kapanadze, Elena Egorova, Nart Kuneshko, Jean-Christophe Gris, Ismail Elalamy, Cihan Ay, Alexander Makatsariya

**Affiliations:** 1Department of Obstetrics, Gynecology and Perinatal Medicine, I.M. Sechenov First Moscow State Medical University (Sechenov University), Trubetskaya Str. 8-2, 119991 Moscow, Russia; alvorobev@gmail.com (A.V.); vikabits@mail.ru (V.B.); fidan.yagubova.2021@gmail.com (F.Y.); totu1@yandex.ru (J.K.); antoninasolopova@yandex.ru (A.S.); tretyakova777@yandex.ru (M.T.); grigkristik96@gmail.com (K.G.); eynullayevas00@gmail.com (S.E.); cherepanov_a_g@staff.sechenov.ru (A.C.); eesdoctor@mail.ru (E.E.); jean.christophe.gris@chu-nimes.fr (J.-C.G.); ismail.elalamy@aphp.fr (I.E.); cihan.ay@meduniwien.ac.at (C.A.); gemostasis@mail.ru (A.M.); 2Faculty of General Medicine, Russian University of Medicine, 4th Dolgorukovskaya Str., 127006 Moscow, Russia; d2608m@gmail.com; 3Faculty of General Medicine, I.M. Sechenov First State Moscow Medical University Baku Branch, Huseyn Javid, Yasamal, Baku AZ1141, Azerbaijan; gadzhievaa.2002@mail.ru; 4Faculty of General Medicine, Pirogov Russian National Research Medical University, Ulitsa Ostrovityanova 1, 117997 Moscow, Russia; emiliakhalilulina@gmail.com; 5Center of Pathology of Pregnancy and Hemostasis «Medlabi», 340112 Tbilisi, Georgia; medlabimedlabi@gmail.com; 6Moscow’s Region Odintsovo Maternity Hospital, 143003 Odintsovo, Russia; drnartfaruk@mail.ru; 7Faculty of Pharmaceutical and Biological Sciences, Montpellier University, 34093 Montpellier, France; 8Department Hematology and Thrombosis Center, Medicine Sorbonne University, 75012 Paris, France; 9Hospital Tenon, 4 Rue de la Chine, 75020 Paris, France; 10Department of Medicine I, Clinical Division of Hematology and Hemostaseology, Medical University of Vienna, 1080 Vienna, Austria

**Keywords:** thrombotic microangiopathy, cancer patients, thrombotic thrombocytopenic purpura, thrombo-inflammation, ADAMTS13, drug-induced TMA, COVID-19, microangiopathic hemolytic anemia, chemotherapy-induced TMA

## Abstract

Thrombotic microangiopathy (TMA) encompasses a range of disorders characterized by blood clotting in small blood vessels, leading to organ damage. It can manifest as various syndromes, including thrombotic thrombocytopenic purpura (TTP), hemolytic-uremic syndrome (HUS), and others, each with distinct causes and pathophysiology. Thrombo-inflammation plays a significant role in TMA pathogenesis: inflammatory mediators induce endothelial injury and activation of platelet and coagulation cascade, contributing to microvascular thrombosis. Primary TMA, such as TTP, is primarily caused by deficient ADAMTS13 metalloproteinase activity, either due to antibody-mediated inhibition or intrinsic enzyme synthesis defects. In cancer patients, a significant reduction in ADAMTS13 levels and a corresponding increase in VWF levels is observed. Chemotherapy further decreased ADAMTS13 levels and increased VWF levels, leading to an elevated VWF/ADAMTS13 ratio and increased thrombotic risk. Drug-induced TMA (DITMA) can result from immune-mediated or non-immune-mediated mechanisms. Severe cases of COVID-19 may lead to a convergence of syndromes, including disseminated intravascular coagulation (DIC), systemic inflammatory response syndrome (SIRS), and TMA. Treatment of TMA involves identifying the underlying cause, implementing therapies to inhibit complement activation, and providing supportive care to manage complications. Plasmapheresis may be beneficial in conditions like TTP. Prompt diagnosis and treatment are crucial to prevent serious complications and improve outcomes.

## 1. Introduction

Thrombotic microangiopathy (TMA) denotes a spectrum of disorders characterized by the occurrence of blood clotting within the small vasculature. This subsequently occurs, affecting the microvessels of certain organs and leading to organ damage. This pathological state precipitates hemolytic phenomena, elevates bilirubin serum levels, and induces thrombocytopenia, which sequentially fosters systemic coagulopathies and hemorrhagic manifestations. TMA epitomizes a rare yet severe clinical entity founded upon systemic microvascular thrombosis. Laboratory diagnostics for this syndrome entail the identification of microangiopathic hemolytic anemia, characterized by red blood cell fragmentation, alongside a concomitant reduction in platelet count. Associated with hemolysis, further laboratory findings include increased levels of lactate dehydrogenase (LDH), reticulocytosis, a deficiency or absence of haptoglobin, and, typically, a surge in unconjugated bilirubin. Histological evaluation reveals systemic thrombus deposition, with a predilection for smaller or sometimes larger vessels, and the composition of these thrombi is diverse, contingent upon the etiology of TMA [1].

TMA may present with a cohort of syndromes and pathologies that include [2,3]:Thrombotic thrombocytopenic purpura (TTP), characterized by reduced ADAMTS13 activity;Hemolytic–uremic syndrome (HUS);Hereditary deficiency of the metalloproteinase ADAMTS13, known as Upshaw–Schuman syndrome;Hereditary or acquired forms of atypical hemolytic-uremic syndrome (aHUS);Hemolytic–uremic syndrome associated with Shiga toxin-producing *Escherichia coli* (STEC-HUS);TMA induced by medications that exert a direct toxic effect upon the endothelium, such as certain chemotherapeutic agents and drug-induced immune-mediated endothelial injury, etc.

The pathophysiological underpinnings of these syndromes differ; however, they are each characterized by the thickening of capillary and arteriole walls. This is accompanied by notable edema and damage to the endothelial cells, as well as the formation of platelet clots that occlude the implicated vessels. Such disruptions in the microcirculatory system on a widespread basis lead to the emergence of ischemia and subsequent infarction within various organs [4]. The renal and central nervous systems are frequently the primary systems afflicted by these pathological states. Conditions such as HUS and TTP are of particular concern, given that without timely diagnosis and treatment, the progression of these diseases may result in fatal outcomes.

## 2. The Historical Trajectory of the Identification and Ensuing Exploration of TMA

Precisely a century prior, in the year 1924, a seminal case report by Dr. Eli Moschkowitz emerged, delineating the inaugural account of a patient who succumbed to an at that time unidentified acute disorder, presently recognized as TTP [5]. Post-mortem examination disclosed the presence of pervasive hyaline microthrombi within the capillaries and arterioles of numerous organs, notably the heart, spleen, and kidneys.

In 1947, Singer and colleagues meticulously documented a case mirroring the pathology of Moshkowitz’s earlier reported patient, with autopsy findings revealing extensive microthrombi present in various organs [6]. After a comprehensive review of all hitherto reported cases exhibiting analogous clinical presentations, they posited the existence of a distinct pathological entity. Singer et al. juxtaposed this condition against the then-termed “idiopathic thrombocytopenic purpura” (currently referred to as immune thrombocytopenia) and recommended the nomenclature “thrombotic thrombocytopenic purpura” for this disease [6].

In their seminal 1966 study, Amorosi and Ultmann conducted a comprehensive review of all patients then registered with a diagnosis of TTP [7]. They elucidated a quintet of diagnostic criteria: microangiopathic hemolytic anemia characterized by red blood cell fragmentation on blood smear analysis, including schistocytes and helmet cells; consumption thrombocytopenia; neurological manifestations; renal impairment; and fever. Amorosi and Ultmann raised inquiries regarding the nature of “hyaline” microthrombi and remarked on the near uniformity of fatal outcomes among the cohort while also underscoring the apparent ineffectiveness of extant therapy [7].

Throughout numerous years, a multitude of scientists have posited assorted conjectures pertaining to the etiology and pathogenesis of TTP [8]. It was not until 1982 that Moake et al. delineated the presence of ultra-large von Willebrand factor (VWF) multimers in individuals diagnosed with chronic relapsing TTP [9]. Moake et al. proffered that these multimers, extant in the plasma during periods of remission, played a pivotal role in the precipitous conglomeration and accretion of platelets within the microvasculature amid the acute phases of TTP, noting their conspicuous absence from the plasma during said episodes.

In the year 1996, two investigative teams reported separately on the identification of a novel metalloprotease extracted from human plasma, noted for its specific degradation of VWF [10,11]. Subsequently, in 2001, three scientific groups accomplished the purification of the aforementioned VWF-cleaving protease to homogeneity, thereby facilitating the examination of its amino acid composition [12,13,14]. Proceeding from this groundwork, Zheng et al. designated the protease as a novel member of the ADAMTS family, categorizing it as ADAMTS13 [15].

In 1959, Rubinstein et al. documented an unanticipated remission in a subject afflicted with TTP following the administration of fresh whole blood [16]. In the years that followed, literature emerged detailing several instances where positive outcomes were associated with interventions such as whole blood exchange transfusion, plasmapheresis with fresh frozen plasma replacement, or the administration of substantial volumes of plasma infusions [17]. Corticosteroids have been employed in the treatment of less severe cases, occasionally obviating the need for plasmapheresis [18]. The fundamental objective of treatment is the suppression of autoantibody production.

## 3. Clinical Variations and the Epidemiological Characteristics of TMA

Notably, the predominant forms of TMA are TTP and HUS, both of which may manifest as acquired or hereditary conditions [19]. A defining characteristic of TTP is the insufficient activity of ADAMTS13—a protease responsible for the cleavage of VWF. HUS, conversely, is frequently instigated by infection with Shiga toxin-producing strains of bacteria, particularly *Escherichia coli* 0157:H7, hence the appellation STEC-HUS. Meanwhile, cases not associated with such infections but instead with anomalies in the alternate complement pathway are categorized as aHUS [1].

TTP is typified by a constellation of clinical manifestations, which include thrombocytopenia, microangiopathic hemolytic anemia (MAHA), fever, renal impairment, and neurologic deficits. HUS similarly exhibits thrombocytopenia and MAHA, yet it is distinguished by a more pronounced renal involvement [20].

Pregnancy may significantly precipitate TMA, necessitating the imperative consideration of microcirculatory disturbances in affected patients. Such circumstances warrant meticulous diagnostic processes to preclude conditions that pose a mortal risk to both the mother and the fetus [21,22,23,24]. Consequent morbidities, including renal insufficiency, seizures, cerebrovascular events, pulmonary congestion, and disseminated intravascular coagulation, are documented possibilities [25]. The under-recognition of TMA’s potential in pregnant women is frequently attributed to the clinical overlap with obstetrical pathologies, notably those related to the placenta. During gestation or in the postpartum period, obstetricians may erroneously diagnose TTP as HELLP syndrome or severe preeclampsia, given the paralleled symptomatology of thrombosis, hemolysis, nephropathy, and elevated blood pressure [26,27]. The distinction between preeclampsia and the rarer TMA variants during gestation remains a diagnostic challenge due to substantial congruence in clinical and laboratory presentations [28]. Given their rarity, these TMA forms lack empirical research that could undergird evidence-based guidelines for their identification and therapeutic approach amid pregnancy.

MAHA emerges as a variant of TMA during the course of chemotherapy. It is distinctively marked by the presence of thrombocytopenia, intravascular coagulation, and ischemic organ damage [29].

Within the wider general population, conditions of TMA, such as HUS with a prevalence of 1–3 cases per 100,000 individuals and TTP with a prevalence of less than 1 per 100,000, are considered relatively infrequent. However, the overall incidence of TMA among oncology patients ranges from 6–15% and can rise to approximately 40% among recipients of hematopoietic stem cell transplantation (HSCT) [30]. A recent retrospective analysis of hospitalized patients diagnosed with TMA conducted by Bayer et al. [31] suggests that secondary TMA predominates over primary TMA, a finding corroborated by subsequent research [32]. While primary TMA typically exhibits hereditary origins, secondary TMA emerges due to a multifactorial etiology encompassing a plethora of causative factors. These include but are not limited to acquired ADAMTS13 deficiency, which may be concomitant with severe pathological states such as severe preeclampsia, premature separation of the normally implanted placenta, septic conditions, systemic inflammatory response syndrome (SIRS), and cancerous processes. Irrespective of the divergent pathogenetic mechanisms, endothelial damage that precipitates microvascular ischemia remains the defining characteristic of TMA [33,34].

In a study conducted within four major healthcare institutions in France, researchers examined the progression of TMA across a cohort of 564 patients over a temporal span extending from 2009 to 2016. The prevalence of primary and secondary TMA, etiological factors contributing to the onset of secondary variants, and primary clinical outcomes during hospitalization—including mortality, the necessity for renal dialysis, and severe cardiovascular events (comprising acute coronary syndromes and/or congestive heart failure)—were evaluated. Primary TMA was identified in a modest fraction of the cohort, representing 33 out of 564 patients (approximately 6%), and was typified by conditions such as TTP and aHUS. In the observed cohort, secondary TMA was diagnosed in 531 out of 564 patients, representing 94% of the sample. Notably, within this subgroup, an etiological factor was delineated in 94% of instances (500 of 531). Within this subset, associations were drawn with pregnancy (35%), oncological malignancies (19%), infections and SIRS (33%), exposure to specific pharmacological agents (26%), organ transplantations (17%), autoimmune disorders (9%), Shiga toxin-producing *Escherichia coli* infections (6%), and malignant hypertension (4%). Throughout the course of hospitalization, renal dialysis was necessitated in 84 patients (15%), severe cardiovascular complications arose in 64 patients (11%), neurological complications occurred in 25 patients (4%), and TMA concluded fatally in 58 patients (10%). Notably, drug-induced TMA (DITMA), encountered in 26% of patients, was predominantly ascribed to the administration of calcineurin inhibitors (68%), gemcitabine (8%), and inhibitors of vascular endothelial growth factor (3%). It merits emphasis that a significant proportion, almost half, of the studied patient cohort had more than two suspected etiological contributors [31].

The incidence of DITMA within the general populace, as well as its role in the etiology of chronic kidney disease, remains poorly characterized, a circumstance attributable to the absence of standardized diagnostic guidelines. An incisive systematic review revealed that a mere 344 out of an estimated 1500 articles concerning DITMA provided data amenable to evaluation; a scant 22 out of 78 distinct drug-TMA correlations were deemed definitive, while a substantial number were downgraded to a status of probable [35]. Notwithstanding these figures, the diagnosis of DITMA is conjectured to be comparatively underrecognized, as it can manifest as a form of TMA with primarily renal impairment, devoid of systemic manifestations that might otherwise aid in the diagnostic process [36,37]. Specifically, the incidence of chemotherapy-related TMA is surmised to be on the rise, owing to the heightened application of multi-agent treatment protocols, and this phenomenon may constitute an insufficiently acknowledged etiology of chronic kidney disease in oncology patients, where hematologic anomalies are frequently ascribed to myelosuppression, thus postponing accurate diagnosis [29].

## 4. Pathogenetic Variants in TMA

TMA exhibits a heterogeneous and frequently obscure pathogenesis with diverse clinical presentations. Renal compromise is frequent, as the kidneys are especially prone to endothelial damage and microvascular thrombosis. Given the significant morbidity and mortality rates linked to TMA, prompt assessment, accurate diagnosis, and the commencement of suitable therapeutic interventions are imperative [38].

### 4.1. Role of ADAMTS13 in the Pathogenesis of TMA

The pivotal element in the etiopathogenesis of TTP is an insufficiency in the enzymatic activity of ADAMTS13 metalloproteinase. This deficiency arises secondary to antibody-mediated inhibition or an inherent defect in the enzyme’s synthesis. Consequently, this impediment leads to the absence of cleavage of VWF multimers, which subsequently engage in the activation of platelet aggregation. This interaction occurs through the binding of VWF to platelet surface glycoproteins. The formation of platelet thrombi within the microcirculation precipitates endothelial damage, characteristic of microangiopathy, as well as the hemolytic destruction of erythrocytes [39]. A number of studies have observed an association between TMA and autoimmune inflammatory conditions, including systemic lupus erythematosus, rheumatoid arthritis, and various glomerulopathies. Furthermore, there is evidence to suggest a correlation between TMA and the post-transplant setting, both in bone marrow and solid organ transplant recipients [33].

### 4.2. Features of Drug-Induced TMA

The initiating incident in TMA is hypothesized to originate from endothelial injury, which may be attributed to the influence of antibodies, the presence of circulating immune complexes, bacterial endotoxins, or pharmacological agents, ultimately leading to the aggregation of platelets and the provocation of the complement system. The underlying mechanisms of DITMA are proposed to be multifactorial. Endothelial cells of the vasculature are responsible for the synthesis of critical components that facilitate coagulation and platelet aggregation, such as VWF, prostacyclin, and thrombomodulin. The equilibrium of these factors is pivotal for maintaining endothelial cell integrity. Notably, a reduction in the concentration of prostacyclin, a powerful inhibitor of platelet aggregation, has been documented in the context of DITMA development, as delineated in reference [29].

Two principal mechanisms have been postulated for DITMA, specifically immune-mediated and non-immune-mediated pathways [40]. The non-immune-mediated process is characterized by either dose-dependent or duration-dependent injury to the endothelium, which precipitates platelet aggregation and subsequent complement system activation. Conversely, the immune-mediated mechanism of DITMA is correlated with the formation of drug-induced antibodies and the propensity of drugs to bind to platelets in a drug-dependent manner, generally manifesting within a temporal window ranging from several hours to 21 days subsequent to drug exposure. Agents such as gemcitabine and oxaliplatin are postulated to induce TMA via both immune complex-related pathways and direct cytotoxic effects, contingent on the timing and the patient’s therapeutic response. Nonetheless, it must be acknowledged that the prevalence of DITMA might be underreported, given the infrequency of genetic assessment in instances where medication is implicated as causative. Specifically, the proteasome inhibitor carfilzomib has been implicated in complement-mediated TMA (CM-HUS) via a reduction in complement factor H (CFH) expression. Within a studied cohort, two out of three patients who developed carfilzomib-induced TMA and were positive for the CFH-related protein 3 also exhibited the CFHR3-CFHR1 gene variant; the implications of this heterozygous variant, however, remain to be fully elucidated [16].

### 4.3. Cancer-Associated TMA

Continued scientific inquiry progressively augments our comprehension of the determinants that govern tumorigenesis and the multifaceted stages within the tumoral cascade. Notwithstanding, the correlation between tumorigenesis and TMA is not uniformly discernible, and the repercussions of diverse tumor-related growth factors on this process can be intricate, as depicted in Figure 1.

TMA within oncological patients manifests as a consequence of multiple etiologies, and the contributory elements to its pathogenesis include the following:Neoplastic microangiopathy: malignancies such as lymphomas, leukemias, metastatic carcinomas, and hemangiopericytomas have the capacity to incite microangiopathic changes, precipitating the coagulation of blood within diminutive vascular structures [41].Contributory neoplastic factors to the genesis of microangiopathy [42] are
A.The direct invasion of adjacent tissues by the tumor potentially resulting in the obstruction of small blood vessels and the subsequent disturbance of vascular perfusion;B.The process of tumor neoangiogenesis;C.The necrosis and degradation of neoplastic tissue, which may actuate the activation of the coagulatory cascade.
2.Disruption of Vascular Endothelium Integrity: neoplastic proliferation and the associated tumor microenvironment are implicated in the impairment or outright disruption of the vascular endothelium. This pathology may subsequently facilitate thrombus formation [43];3.Tumor-Induced Hypoxia and Necrosis: within the tumor matrix, hypoxic conditions often prevail, potentially correlating with microthrombi development and coagulatory system activation [44];4.Hemostatic System Activation: malignant neoplasms, contingent upon their histological classification, can initiate various degrees of coagulation pathway activation, thereby promoting thrombogenesis. Oncogenic factors and molecules associated with tumor presence may have interactions with the coagulatory cascade, amplifying the propensity for TMA;5.Paraneoplastic Syndromes: certain neoplasms may precipitate paraneoplastic syndromes that subsequently impact the vascular system and coagulation pathways.A.Disseminated Intravascular Coagulation (DIC) syndrome may manifest as a sequela of various malignancies, notably those originating from the stomach, pulmonary, prostatic, and additional organ systems. This syndrome is associated with compromised platelet functionality and the genesis of microthrombi within diminutive vessels, potentially culminating in the onset of TMA;B.Polycythemia Vera (PV) is distinguished by a hyperproliferation of erythrocytes within the bone marrow and may present concomitantly with thrombocytosis and amplified thrombotic risk. TMA may emerge as a clinical presentation within this disorder;C.Goodpasture’s Syndrome, an infrequent paraneoplastic condition frequently linked to colorectal cancer, is typified by a constellation of findings, including TMA, hyperproteinemia, and renal pathology [45].

Furthermore, the neoplasm itself may induce thrombocytopenia owing to systemic microvascular metastases; obstruction of microvasculature by neoplastic cells leads to erythrocyte fragmentation and platelet consumption within tumor emboli. Thrombocytopenia may also result from extensive metastatic infiltration of the bone marrow or consequent necrosis. These conditions are predominantly observed in malignancies of the stomach, lung, breast, and prostate, particularly in adenocarcinomas, as well as in hematologic malignancies such as lymphomas, which constitute approximately 8% of cases. Cancer-associated TMA tends to present with bone pain more frequently than TTP and often demonstrates an insufficient response to plasmapheresis.

## 5. The Diagnostic Criteria for TMA

TMA presents as a multifaceted issue in terms of diagnosis and treatment. The algorithm employed for the diagnosis of TMA entails not only the initial identification but also a subsequent delineation between the primary and secondary subtypes of TMA. Within the domain of primary TMA, it is imperative to discern between infection-induced HUS (which arises following exposure to Shiga toxin-producing pathogens such as enterohemorrhagic or enteroaggregative *Escherichia coli*, or *Shigella dysenteriae* type I [STEC-HUS]), aHUS, and TTP [46].

Patients with TMA typically present with the following clinical manifestations [47]:gastrointestinal symptoms such as nausea, vomiting, abdominal pain, and altered bowel movements, including frequent watery stools, which may occasionally be blood-stained;fluid retention presenting as peripheral and cavity edema, oliguria or anuria, and discoloration of urine;respiratory distress evidenced by dyspnea;dermatological signs, including hemorrhagic rashes;neurological complaints like headaches and visual disturbances, which may range from slight blurring to complete loss of vision;general symptoms including malaise, fatigue, anorexia, vertigo;neurobehavioral changes such as psychomotor fluctuations or seizures and partial paralysis.

Laboratory Research includes the following:Complete Blood Count: evaluation includes checking for anemia (specifically microangiopathic hemolytic anemia (MAHA) that is Coombs-negative), thrombocytopenia (platelet count under 150 × 10^9^/L or a decline exceeding 25% from the baseline), and reticulocytosis (notably, MAHA can occasionally manifest in the absence of thrombocytopenia);Peripheral Blood Smear: investigation for schizocytes, where the criterion is an excess of 0.1% in a peripheral blood smear;Biochemical Blood Test: measurements include lactate dehydrogenase (LDH) levels, which should be elevated, and/or haptoglobin levels, which should be reduced. The manifestation of acute kidney injury is characterized by elevated concentrations of creatinine and urea, along with disturbances in electrolyte balance such as hyperkalemia, hyponatremia, hypocalcemia, and hyperphosphatemia. Concurrently, there is an increase in uric acid levels, indicative of hyperuricemia. In cases where nephrotic syndrome ensues, significant reductions in protein levels are observed, notably hypoproteinemia and hypoalbuminemia, accompanied by an increase in α2-globulin. Additionally, this condition is associated with heightened levels of cholesterol and triglycerides, denoting hypercholesterolemia and hypertriglyceridemia, respectively.General Urine Analysis: may reveal proteinuria, with the potential presence of either microhematuria or macrohematuria;Antiglobulin Test: should demonstrate a negative Coombs test;Complement System Components Assessment: a decline in C3 concentrations, occurring simultaneously with consistent C4 levels, can be observed in roughly 50% of patients with aHUS. It is important to note that standard C3 levels do not rule out aHUS. In patients with STEC-HUS, a decline in C3 levels might occur, akin to any other infection, yet C4 levels may stay within the normal range [38].

MAHA concurrent with thrombocytopenia is indicative of thrombotic microangiopathy, which is characterized by the formation of thrombi within small or larger vascular structures [48]. In oncology patients, pervasive microvascular metastatic dissemination or pronounced bone marrow impairment may lead to thrombocytopenia. A clinical manifestation reminiscent of DIC syndrome may arise secondary to septic conditions or the neoplasm itself. Moreover, antineoplastic treatments have the potential to induce TMA or to instigate dose-dependent toxicities, including idiosyncratic immune-mediated responses precipitated by drug-dependent antibodies. It is noteworthy that numerous etiologies of TMA observed in oncology patients are not amenable to treatment via plasmapheresis. Consequently, where feasible, addressing the underlying malignancy is imperative for managing both TMA and DIC concomitantly. It is essential to recognize DITMA as a potential complication and ensue with the cessation of the implicated pharmacological agent [48].

**Biomarkers.** Clinical manifestations suggestive of intravascular hemolysis may be absent in instances of mild systemic hemolysis or in TMA that is predominantly renal. In this regard, the examination of distinct biomarkers emerges as a critical component in facilitating prompt diagnosis [49]. Within a cohort consisting of 39 individuals suffering from HUS linked to the presence of Shiga toxin-producing *Escherichia coli*, the serum neutrophil gelatinase-associated lipocalin was scrutinized for its value as a prognostic instrument pertaining to the onset of renal replacement therapy [50]. Notwithstanding, there appears to be a more pronounced utility for endothelial markers when compared to tubular injury markers in the context of TMA. Notably, escalated levels of endothelial markers, exemplified by thrombomodulin, plasminogen activator inhibitor-1, and the soluble form of intercellular adhesion molecule-1, have been documented in cases of transplant-associated TMA. In parallel, heightened levels of thrombomodulin, tissue plasminogen activator, and plasminogen activator inhibitor-1 have been identified in TMA associated with mitomycin-C. The deployment of endothelial biomarkers in diagnosing other variants of TMA warrants additional investigative endeavors to establish their diagnostic efficacy [29].

The clinical manifestations of the syndrome are outlined [1] as follows:the presentation of Coombs-negative MAHA, which is characterized by elevated serum LDH levels, serum haptoglobin concentrations that are undetectable or markedly diminished, and the presence of schistocytes in the peripheral blood smear—a condition that is not, however, a compulsory criterion;he occurrence of thrombocytopenia;the involvement of multiple organ systems, notably characterized by renal dysfunction, neurologic anomalies, and gastrointestinal symptoms; renal complications may present as acute renal injury, proteinuria, or hypertension;the maintenance of normal coagulation profiles.

Pulmonary manifestations are rare in instances of TTP, but they concurrently appear in over 70% of oncological TMA, which are associated with DIC. Abnormal liver function tests and moderate to severe impairment of renal function are commonly noted in the presence of TMA concomitant with neoplastic processes [51].

## 6. The Significance of Assessing ADAMTS13 and VWF in Patients with Gynecological Cancers

TTP is precipitated by a severe deficiency in ADAMTS13, culminating in the sustained presence of ultra-large VWF multimers, which consequently initiates the formation of microthrombi [52]. The majority of TTP occurrences are attributable to immune-mediated mechanisms. Nonetheless, a hereditary variant of the condition does manifest, stemming from homozygous mutations within the ADAMTS13 gene [53].

ADAMTS13 functions as a metalloproteinase integral to the coagulation system [54]. Its principal role involves the cleavage of VWF fragments into the bloodstream, thus regulating VWF activity by cleaving the ultra-large VWF multimers into diminished, less active units [55]. A reduction in ADAMTS13 activity, coupled with an elevated VWF concentration, may act as a harbinger of microcirculatory disturbances, which hold significant implications in the pathogenesis of multiple organ dysfunction [55].

VWF, a multifaceted glycoprotein, plays a pivotal role in hemostasis, primarily facilitating platelet adhesion to compromised vascular sites, thereby catalyzing the incipient phase of thrombus generation [56]. Furthermore, VWF contributes to the stabilization of factor VIII within the circulatory system, thus extending its functional duration and enhancing coagulative efficacy [57].

Additionally, VWF is implicated in the regulation of angiogenesis, supervising vascular proliferation while also exerting influence on inflammatory mechanisms and immunological functions [58].

Pathological states may arise from either a deficiency or an overabundance of VWF, resulting in clinical manifestations ranging from thrombotic events to hemorrhagic conditions and potentially autoimmune disorders. A critical component in modulating VWF activity is the enzymatic action of ADAMTS13, which is instrumental in preserving hemostatic equilibrium and precluding undue platelet activation and subsequent thrombus consolidation.

In the present cohort-controlled study, an examination of 108 women diagnosed with ovarian, cervical, and breast cancer was undertaken. These patients were bifurcated into two distinct groups: Group I comprised 48 cancer patients with thrombotic episodes in history, and Group II constituted 60 cancer patients devoid of clinically significant thrombotic manifestations. Additionally, a Control group of 25 women, absent of malignant neoplasms, was scrutinized. The plasma levels of ADAMTS13 and VWF were quantified both pre- and post-chemotherapy.

The findings evidenced a statistically significant disparity in the levels of ADAMTS13 and VWF between the patients with oncological conditions from Groups I and II when juxtaposed with the Control group. Specifically, in Group I, the ADAMTS13 level was 1188–1317 IU/L, markedly reduced (*p* < 0.01) in comparison to Group II, which was 1402–1511 IU/L, and the Control’s level of 1572 IU/L. The VWF levels in Group I were recorded at 1763–1892 IU/L, substantially elevated relative to Group II (1272–1428 IU/L) and the Control group (1014 IU/L). Subsequent to chemotherapy, there was a discernible decline in ADAMTS13 levels within Groups I and II: the concentrations diminished to 943–1021 IU/L and 1337–1358 IU/L, correspondingly. Concurrently, there was an ascension in VWF levels in both cohorts to 1878–1923 IU/L for Group I and 1446–1513 IU/L for Group II.

In the preponderance of cases encompassing subjects with an antecedent of thrombotic events, the parameters under investigation were within or marginally deviated from established reference values. Consequently, for the purpose of enhancing diagnostic precision, the integral indicator, consisting of the VWF to ADAMTS13 ratio, was computed. It was through this computation that the most salient alterations in the hemostatic profile were discerned, exhibiting a correlation with the levels of the DIC syndrome biomarker, D-dimer.

Hence, it is the VWF/ADAMTS13 ratio that proves to be pivotal in the stratification of thrombotic risk and serves as an adjunctive marker, in conjunction with D-dimer levels, in the prescription of anticoagulant regimes. This ratio further acts as a criterion for gauging the efficacy of antithrombotic prophylaxis.

Although the study cohort and the Control group both exhibited standard levels of ADAMTS13 activity throughout the chemotherapy treatment, a trend towards a further decrement was observed; the activity of metalloproteinase remained within the normal reference range. Concurrently, there was an elevation in the levels of VWF and ADAMTS13 inhibitors during the chemotherapy sessions. This culminated in a marked and successive elevation in the VWF/ADAMTS13 ratio.

In the patient cohort undergoing chemotherapy, it was observed that the concentration and activity of ADAMTS13 diminished, while, concomitantly, there was a rise in ADAMTS13 inhibitors due to chemotherapeutic toxicity impacting the endometrial layer. Additionally, an elevation in VWF levels was noted. Consequently, these alterations culminated in an enhanced potential for hyperaggregation within the circulatory system. This was accompanied by a marked reduction in the physiological compensatory mechanisms for ULWVF multimer degradation, thereby significantly escalating the thrombotic risk.

The present study revealed a correlation between the diminished level and activity of ADAMTS13 and the augmented level of VWF across the sample cohort. This correlation could be attributable to impaired synthesis of metalloproteinase within hepatic tissues or to endothelial dysfunction, which may be a sequela of invasive neoplastic growth or an adverse effect of chemotherapeutic interventions.

The current study delineates that individuals diagnosed with malignant neoplasms of the female reproductive tract exhibit a pronounced activation of the hemostatic system. The findings elucidate that there is a direct correlation between the extent of tumor dissemination and the severity of hemostatic system dysfunction, which intrinsically elevates the risk of thrombotic events. This correlation is corroborated by a multitude of antecedent studies.

In the context of carcinoma of the cervical canal (adenocarcinoma) classified under stages I–III, the probability of thrombotic complication emergence is comparatively lower than that observed in ovarian or breast cancers. This disparity is attributable to the distinct methodology employed in staging the progression of cervical cancer. Moreover, it is observed that ovarian cancer patients are more prone to developing DIC syndrome. This increased incidence can be primarily ascribed to the substantial morphological heterogeneity of the ovarian tissue and, secondarily, to the secretion of a mucinous constituent by a predominant number of ovarian tumors.

## 7. Differential Diagnosis of TMA

TMA in oncological patients may be attributable to the intrinsic neoplastic process or therapeutic interventions against cancer, or it may represent an independent pathological entity. It is imperative to discern between TTP and aHUS in the presence of thrombocytopenia, as swift differentiation is crucial; these conditions necessitate divergent therapeutic approaches, and the expeditious commencement of appropriate therapy significantly influences patient prognosis (Figure 2).

HUS typically arises following an intestinal infection with Shiga toxin-producing microorganisms, notably *Escherichia coli* 0157 [59]. Predominantly observed in pediatric cohorts, these patients often present with initial abdominal pain and the archetypical manifestation of bloody diarrhea [60].

Furthermore, CM-HUS constitutes a particularly uncommon yet significant etiology of TMA that warrants recognition owing to its amelioration upon complement inhibition [61].

## 8. Thrombo-Inflammation and TMA

Thrombo-inflammation refers to a pathological state characterized by concomitant vascular inflammation and the formation of thrombi. This condition may manifest within any vessel and is frequently linked with a multitude of pathologies, including infectious diseases, neoplastic disorders, and autoimmune conditions. The emergence of thrombo-inflammation can precipitate grave sequelae like myocardial infarction, cerebrovascular accidents, and deep vein thrombosis. TMA is distinguished by the presence of microthrombi within the microvasculature, precipitating circulatory compromise and subsequent organ dysfunction. The inflammatory cascade is integral to the pathogenesis of the aforementioned condition, with the etiology often rooted in various processes, encompassing the activation of the immune system, tissue damage, and the liberation of pro-inflammatory cytokines [62].

Pro-inflammatory mediators induce endothelial injury, thereby rendering the procoagulant constituents of the subendothelial matrix accessible to platelets. This matrix harbors multiple elements, such as collagen, VWF, vitronectin, and others, that recruit and subsequently activate platelets. VWF engages platelets through its binding to the glycoprotein Ib receptor on the platelet surface, facilitating their adherence to the impaired endothelium. Moreover, VWF acts as an acute phase reactant during inflammatory states and serves as an indicator of endothelial perturbation amid such conditions.

During inflammatory states, the activation of platelets can proceed under the auspices of pro-inflammatory mediators, notably the platelet-activating factor (PAF). Concomitantly, systemic activation of the coagulation cascade contributes to the production of copious thrombin, which acts as a potent inducer of platelet activation. Upon exposure to endotoxins and cytokines, platelets undergo activation, eliciting the release of aggregation-promoting substances, including epinephrine, adenosine diphosphate (ADP), serotonin, and thromboxane A2. This results in substantial platelet clumping within the vasculature, leading to their subsequent disintegration and the liberation of aggregation stimulants into the plasma. Subsequently, this phenomenon may precipitate into hyperaggregation of platelets.

The regulatory mechanisms governing coagulation and inflammation are significantly influenced by the interactions between platelets and leukocytes. Platelets in an activated state synthesize P-selectins on their surface, which facilitate adhesion to monocytes, polymorphonuclear leukocytes (PMNs), and select T lymphocyte populations. Furthermore, platelets engage leukocytes via P-selectin and the PSGL-1 receptor located on leukocytes, bolstering their recruitment to the vascular endothelium and their involvement in the inflammatory cascade.

Upon the onset of hemorrhage, there ensues an interaction between leukocytes and thrombocytes, characterized by adhesion, activation, and subsequent robust amalgamation. Platelet P-selectins engage with PSGL-1 receptors situated on the leukocyte surface, facilitating the transposition of leukocytes to the vascular wall. This interaction is pivotal for the incorporation of leukocytes within thrombotic structures and their transmigration across the endothelial layer. Moreover, P-selectins contribute to intracellular signal transduction and modulate the expression of numerous inflammatory mediators. The consequent adherence of activated leukocytes to thrombocytes augments the production of bioactive substances, including cytokines and coagulation factors, which in turn activate the hemostatic system and the inflammatory cascade.

During the inflammatory response, leukocytes engage in interactions not solely with endothelial cells but amongst themselves as well. This intraleukocytic communication is mediated via L-selectins expressed on the leukocytes and PSGL-1 receptors. It facilitates the adherence of leukocytes within the vasculature to engage in reciprocal signaling, thereby augmenting the mobilization of additional leukocytes to the locus of inflammation [63].

Empirical evidence suggests that concomitant with the inflammatory response, there is an elevation in the concentration of VWF, a substance liberated from activated platelets and endothelial cells, in individuals with prostate cancer subsequent to surgical intervention. Notably, it is observed that the deficiency or inhibition of androgen receptor functionality within neoplastic cells is associated with an augmented propensity for thrombogenesis. This observation implies a potential causative role for the absence of androgen receptors in the enhanced thrombotic risk. In contradistinction, neoplastic cells of the prostate that are characterized by an abundance of androgen receptors do not seem to provoke a similar prothrombotic effect [64].

Inflammatory mediators such as interleukin-8 (IL-8) and tumor necrosis factor-alpha (TNF-α) promote the synthesis of VWF multimers, whereas interleukin-6 (IL-6) and antimicrobial peptides secreted by neutrophils serve to inhibit the activity of ADAMTS13 [65]. Furthermore, in the context of inflammation, both VWF and ADAMTS13 undergo oxidative modifications, culminating in the formation of VWF multimers that are resistant to degradation by the metalloproteinase [66].

Consequently, the dynamics of thrombus inflammation and TMA represent two interrelated pathophysiological states implicated in the genesis of blood clots, particularly within the microvasculature, with the potential to instigate extensive systemic organ damage.

The association between thrombo-inflammation and TMA is delineated by the dysregulation of the hemostatic system, which assumes a procoagulant bias. Variations in the etiological mechanisms are contingent upon the underlying disease or clinical context. It is imperative to acknowledge that both pathologies may precipitate grave sequelae and necessitate prompt therapeutic intervention. The interrelation of thrombo-inflammation and TMA epitomizes a pathological continuum within the ‘reinforcement loop’; herein, thrombo-inflammation precipitates and perpetuates microangiopathic processes; concurrently, TMA augments and propagates inflammatory responses.

## 9. COVID-19 Associated TMA

Within the context of coronavirus infection, particularly in those patients manifesting severe symptoms, there has been the identification of a highly deleterious convergence of three potentially lethal syndromes: DIC syndrome, SIRS, and TMA [67].

Moreover, two instances of TMA concomitant with COVID-19 have been reported. Polymerase Chain Reaction (PCR) assays for SARS-CoV-2 returned positive results in both instances, and the accompanying laboratory findings revealed the presence of MAHA, thrombocytopenia, and acute renal impairment. Furthermore, in neither case were any pre-existing pathogenic mechanisms for TMA development observed [68].

Patients exhibiting severe manifestations of COVID-19 might, as the disease advanced clinically, experience TMA. An elderly patient, aged 78, presented at the hospital exhibiting respiratory distress and diarrhea, symptoms indicative of COVID-19. Following initial improvement, the patient was discharged to continue outpatient care. Subsequently, he was readmitted with clinical signs of acute renal impairment, evidenced by mild anemia and thrombocytopenia. Histological examination of a renal biopsy disclosed the presence of thrombi within the glomerular capillaries, acute tubular necrosis, and increased thickness of the walls of extraglomerular blood vessels, along with C3 deposits observed in the glomerular tufts. Therefore, it is recognized that acute renal dysfunction can manifest as a consequence of secondary TMA, a phenomenon particularly associated with the convalescent phase of coronavirus infections, as referenced in study [69].

In the most critical manifestations of COVID-19, cytokine storm-induced SIRS is prevalent, concomitant with extensive multi-organ damage that may culminate in a fatality. The cytokine storm typifies a self-propagating cascade of pro-inflammatory cytokine release, which has been implicated as a pivotal contributor to multi-organ dysfunction syndrome across various pathological states [70,71,72]. This hyperinflammatory state precipitates a heightened incidence of thrombotic events, subsequently leading to increased mortality in septic patients [73,74]. Despite the primary association of COVID-19 with pulmonary symptomatology, it has been documented that the pathogenesis extends beyond the respiratory system. Indeed, there is evidence to suggest that the virus induces damage systemically, including within the cardiovascular system, where it prompts coagulopathies that potentiate thrombotic episodes. Such episodes predominantly affect the arteries/arterioles, microvasculature, and the venous system, substantially elevating mortality risk [75].

Macro- and microthrombotic complications are frequently observed in the progression of COVID-19. The thrombotic process associated with COVID-19 may involve a confluence of etiological factors, including but not limited to a cytokine storm, the emergence of disseminated intravascular coagulation syndrome, manifestations of antiphospholipid syndrome, TMA, extensive neutrophil extracellular trap (NET) formation, activation of the complement cascade, and diminished fibrinolytic activity.

Thrombocytopathy is a notable manifestation of COVID-19, encompassing both a reduction in platelet count (thrombocytopenia) and an increase in platelet activation, which contribute to a heightened state of coagulability and impairments in immune system functionality. Thrombocytopenia may arise from a decrement in platelet production or from augmented consumption within an expanding thrombus, as well as from escalated rates of platelet apoptosis. Furthermore, there have been reports of autoantibody generation induced by the SARS-CoV-2 virus, targeting platelet surface antigens [76].

Platelet apoptosis is characterized by the secretion of copious pro-inflammatory and procoagulant mediators. Additionally, the SARS-CoV-2 virus has the capacity to form immunological complexes [77]. This virus-induced activation of platelets may result in an augmented formation of platelet-leukocyte aggregates [78], which potentially leads to the liberation of neutrophil extracellular traps (NETs).

Under the pathological milieu of COVID-19, NETs are generated in significant quantities and function to activate the procoagulant pathway in several manners, interfere with fibrinolytic mechanisms, and inhibit the efficacy of anticoagulants [79]. Endothelial cells become activated and may undergo apoptosis due to the cytotoxic impact of histones, the proteinaceous components of NETs. Concurrently, the production of H_2_O_2_ occurs, which further promotes the process of NETosis [80,81].

Moreover, Weibel–Palade bodies, which are resident within the endothelial cells, undergo exocytosis concomitantly with VWF. The subsequent binding of VWF to platelets plays a supportive role in thrombotic events.

NETs have been shown to attenuate the activity of the metalloproteinase ADAMTS13. The components of NETs, including extracellular DNA and histones, possess the ability to adhere to VWF, which serves to amplify the recruitment of additional neutrophils to the inflammatory locale, thereby intensifying the inflammatory response. VWF is integral to the transportation of platelets to areas of vascular injury, facilitating their activation and subsequent aggregation [82]. The endothelial cells’ release of ultra-large VWF multimers can induce the autonomous activation of both platelets and other hematologic cells, augmenting the risk of thrombotic events [83].

Moreover, the pathological processes observed in coronavirus infections and aHUS suggest an implicatory role of the complement system in the etiology of COVID-19, with specific emphasis on the anaphylatoxins C3a and C5a. These molecules are liberated following the enzymatic splitting of C3 and C5, respectively. C5a, in particular, is a significant chemotactic protein capable of provoking the so-called ‘cytokine storm’ mere hours post-infection and instigating the innate immune response. Nevertheless, the overabundance of C5a may precipitate the formation of a pro-inflammatory milieu that is propelled by various mechanisms, contributing to pulmonary damage, lymphocytic depletion, and immune paralysis. The mortality associated with COVID-19 frequently correlates with SIRS, which is characterized by a hypercoagulable state and organ impairment, most notably presenting as microvascular thrombosis in pulmonary and renal structures. In essence, TMA constitutes the mechanism underlying these lethal complications [84].

## 10. The Phenomenon of DITMA

The chemotherapeutic agents most frequently implicated in TMA are mitomycin-C and gemcitabine. Mitomycin-C is a quinine-derived antineoplastic agent obtained from *Streptomyces caespitosus*, known to impede DNA synthesis via the cross-linking of the nucleobases adenine and cytosine [35]. Mitomycin-C exerts a direct toxic effect on endothelial cells and induces platelet aggregation, a process that is inhibited by prostacyclin [85]. The incidence of TMA in patients receiving mitomycin-C ranges from 4% to 15%, with mortality rates reaching up to 75%, contingent upon the overall status of the cancer condition. Symptom onset is classically documented within a four-week period subsequent to the final administered dose. Nonetheless, instances of delayed onset, transpiring 6 to 12 months following the initiation of treatment, have been recorded [86]. In 1989, an analysis conducted using a national registry comprised of 85 American patients with the so-called cancer-associated HUS revealed that 84 of these patients received mitomycin-C treatment, with the majority (75) having been administered doses in excess of 60 mg. It appears that a cumulative dose exceeding 40–60 mg is a significant risk factor [87]. The prevailing toxicity, delayed bone marrow suppression, may obscure the early detection of TMA. Dyspnea, especially notable in noncardiogenic pulmonary edema, was a symptom exhibited by two-thirds of the 84 patients who endured mitomycin-associated TMA. Severe renal failure necessitating dialysis assistance was reported in approximately 33% of the cohort, with roughly 36% achieving recovery [87].

Gemcitabine, a pyrimidine analog, induces apoptosis in rapidly proliferating cells by disrupting DNA synthesis. Predominantly excreted by the renal pathway (98%), the elimination half-life of this compound is contingent upon the infusion period, varying from approximately 0.7 h for infusions under 70 min to an extended 10.6 h for those of the same duration. The inaugural association between TMA and gemcitabine administration was documented in 1994 during a phase II trial concerning pancreatic adenocarcinoma [88]. Further documentation has led to the estimation of its incidence to be between 0.015% and 1.4%. Notably, cumulative doses exceeding 20,000 mg/m^2^ amplify the risk for TMA, yet instances have been reported following the administration of singular or minimal doses, particularly in combination with other agents posing an attendant risk. Prior administration of mitomycin-C has been recognized as a contributing risk factor. Gemcitabine may precipitate both dose-dependent and immune-mediated adverse events [89]. Typically, there is a significant temporal gap between the commencement of TMA and the onset of initial drug effects. The emergence of TMA subsequent to initial gemcitabine administration has been documented within a temporal span ranging from mere days up to 34 months [90]. Additionally, there are other chemotherapeutic agents implicated in the genesis of TMA, albeit with a lower observed incidence.

Analogous to mitomycin, bleomycin operates as an antineoplastic antibiotic that disrupts the synthesis of DNA and exhibits a lower association with TMA. Instances of severe TMA necessitating dialysis, characterized by high mortality rates and diminished reversibility, have been documented in association with bleomycin-induced TMA. However, discerning the causative agent is challenging, as the majority of cases (14 out of 15) were co-administered with cisplatin and/or a vinca alkaloid [91].

Cisplatin, oxaliplatin, and carboplatin have been linked to diverse extents of nephrotoxicity, electrolyte imbalances, and, in exceptional situations, TMA [92]. These agents exert their cytotoxic effects by binding to proteins and nucleic acids, thereby inhibiting DNA replication, inducing cell cycle arrest, and leading to the depletion of ATP. Notably, the induction of TMA has been reported to occur within several hours subsequent to the administration of oxaliplatin, whether as a monotherapy or in conjunction with other pharmaceutical agents like gemcitabine and bleomycin [93]. Typically, renal function impairment observed in these cases was either wholly or partially resolved within a matter of weeks.

Moreover, a limited number of case reports have implicated additional chemotherapeutic drugs, such as vincristine, adriamycin, and 5-fluorouracil, in the occurrence of TMA, albeit predominantly in scenarios involving simultaneous or recent administration with the aforementioned compounds or in conjunction with antiangiogenic therapy [94].

Consequently, DITMA constitutes a specific form of secondary TMA induced predominantly by certain pharmacological agents, frequently those utilized in antineoplastic treatments. The pharmacotherapeutics most commonly implicated in this disorder comprise select immunosuppressive medications, antiangiogenic agents, monoclonal antibodies, and antiplasmin drugs. The clinical manifestations of DITMA include, but are not limited to, hematological abnormalities such as anemia and thrombocytopenia, renal impairment, and neurological disturbances, which may present as headaches, diminished visual acuity, and seizures.

The therapeutic intervention for DITMA primarily entails the cessation of the causative medication coupled with the administration of immunosuppressive treatment. In cases of considerable severity, therapeutic plasma exchange or renal dialysis may be necessitated.

It is critical to acknowledge that whilst the incidence of this condition may be relatively infrequent, the resultant complications are severe and have the potential to be fatal. Vigilance regarding the emergence of DITMA in patients undergoing anticancer therapy is paramount, with a recommendation for diligent monitoring for signs indicative of this condition.

## 11. Treatment Principles for TMA

The treatment of TMA typically involves a multifaceted and personalized approach. Central tenets of intervention entail the following:Determining the etiology of secondary TMA and mitigating or eradication of the inciting factor;Implementation of plasmapheresis, a process designed to extract immunologically active substances from the plasma, which may arrest the advancement of TMA and avert life-threatening sequelae. This modality is particularly efficacious in cases of TTP;Administration of immunosuppressive agents such as glucocorticoids, cyclophosphamide, or rituximab, with the intention of inhibiting the immune system to obviate further progression of TMA;Provision of adjunctive care: TMA can precipitate a multitude of complications, including but not limited to renal impairments, central nervous system disturbances, and anemia. Indispensable supportive therapies, potentially encompassing dialysis, antihypertensive agents, and transfusion of blood products, may be requisite.

An expedited diagnostic assessment is imperative, as the initiation of specific treatment regimens may require immediate attention. For instance, TTP has been associated with a mortality rate of approximately 90% in affected individuals who do not receive prompt pathogenetic intervention. Nonetheless, the amelioration of the fundamental deficiency of the enzyme ADAMTS13 is crucial for averting lethal outcomes. Similarly, the evaluation for CM-HUS is necessary, and upon suspicion, it should be promptly addressed with complement inhibition strategies [95]. Before the introduction of pharmacological complement blockade, the common strategy for treating CM-HUS was therapeutic plasma exchange (TPE). Contrary to TTP, TPE only proves effective for 30% of patients with CM-HUS. Even when there is a response from the hematologic aspect, it does not substantially enhance organ response and overall outcomes [96].

A therapeutic strategy for handling DITMA is emphasized, focusing on the significance of withdrawing medication in conjunction with utilizing therapeutic plasma exchange, rituximab, and anti-complement therapy [97].

Eculizumab represents the primary targeted therapeutic agent for aHUS [98]. This agent is a recombinant humanized monoclonal antibody that selectively binds to complement protein C5, thereby hindering its cleavage into C5a and C5b and mitigating the progression of the disease [99]. Its introduction has been a pivotal development in the management of patients with complement-mediated aHUS and has been deemed to possess a favorable safety profile, even during pregnancy [61].

Individuals receiving eculizumab treatment exhibit increased susceptibility to infections by encapsulated organisms due to the reliance of host defense mechanisms on the membrane complement attack complex. Consequently, immunization against Neisseria meningitidis is imperative, alongside the administration of prophylactic antibiotics for patients undergoing eculizumab therapy [99].

Ravulizumab represents the second therapeutic to receive authorization from the European Medicines Agency and the United States Food and Drug Administration for the management of aHUS. Derived from eculizumab, ravulizumab targets an identical epitope on the C5 component [100]. The engineering of ravulizumab involved histidine substitution in the complementarity-determining regions of eculizumab in maintaining C5 binding in serum while enabling C5 detachment from ravulizumab in the acidified endosomal environment [101]. Moreover, alterations to the Fc region amino acids in eculizumab have enhanced neonatal Fc receptor-mediated recycling efficiency. These modifications have elongated ravulizumab’s serum half-life to approximately 52 days, compared to eculizumab’s 11 days, thereby extending the dosing interval to every 8 weeks as opposed to the biweekly regimen required with eculizumab [102]. It should be taken into account to administer C5 inhibitor as a treatment for any patients who show clinical symptoms suggestive of CM-HUS. Observationally, the long-term outcomes for the kidney have been shown to be better in those cases where eculizumab was started within the first week after the disease appeared [103]. The most effective period for the C5 inhibitor treatment remains a topic of uncertainty. However, approximately half of the patients who stop taking Eculizumab experience a relapse [104]. The most powerful predictor of relapse after ceasing intake is if complement gene variations are detected. Evidence from research, specifically a study involving 38 patients suffering from CM-HUS, showed a relapse in 50% of the patients carrying the MCP mutation and 72% carrying the FH mutation after they stopped their course of eculizumab [105].

To date, there exists a lack of direct comparative analyses between ravulizumab and eculizumab for the treatment of aHUS in studies encompassing adult or pediatric cohorts. Such an absence of comparative data has engendered concerns regarding the equivalent therapeutic efficacy of ravulizumab, as evidenced by a predominance of mortalities and a reduced cessation of dialysis in the adult ravulizumab cohort when contrasted with historical data from eculizumab trials. Notably, the exceedingly minimal incidence of mutation/factor H autoantibodies within the current study intimates that a significant proportion of participants may not have presented with complement-mediated aHUS. Corroborative of this postulation, an analysis of the documented mortalities elucidated that the causative pathology was not complement-mediated aHUS (e.g., instances of sepsis). On an optimistic note, within pediatric investigations where the differential diagnosis for complement-mediated aHUS was more narrowly defined, there was a heightened detection of mutation/factor H autoantibodies, and an impressive 94.4% of the cohort achieved a complete thrombotic microangiopathy (TMA) response by the fiftieth week [38].

Caplacizumab is an innovative single-chain humanized monoclonal antibody developed against the A1 domain of human VWF [106]. This pharmacological agent introduces a unique mode of action, which is centered around the direct prevention of platelet glycoprotein (GP) Ib–VWF association. Evidence gathered from meticulously conducted double-blind clinical studies illustrates its effectiveness in attenuating the fatality rate among iTTP patients [107].

Accordingly, it can be discerned that TMA is a multifaceted condition, the pathogenesis of which may not be unequivocally ascertainable. Consequently, the management of TMA necessitates a tailored therapeutic paradigm, engaging a multidisciplinary team comprising medical professionals from an array of specializations.

At present, no specific drugs have been endorsed for the treatment of TTP. Fresh frozen plasma (FFP) or solvent/detergent-treated plasma infusions are often used to manage acute TTP. There is evidence that some intermediate-purity FVIII:VWF concentrates containing minimal ADAMTS-13 levels have been leveraged as alternative treatments in certain patient groups [108]. Even though plasma infusions can effectively manage TTP, this treatment can often lead to complications such as allergic and anaphylactic reactions or volume overload. For this reason, the creation of a recombinant ADAMTS-13 offers a promising new treatment option that could enhance the current standard of care. The utilization of recombinant ADAMTS-13 introduces a fresh approach to handling cTTP therapy, mitigates numerous hazards linked to human plasma products, and advances the development of highly accurate and personalized dosing schedules [109]. No neutralizing antibodies were observed in a preliminary investigation leveraging this particular therapeutic approach. When used in patients with congenital TTP, recombinant ADAMTS13 was able to augment ADAMTS13 activity to near-normal levels. Associated adverse incidents were typically of mild to moderate intensity, and instances of TTP were infrequent [110].

Following an episode of TMA, patient prognoses can diverge substantially based on the severity of the initial presentation and the timeliness of therapeutic intervention. Chronic outcomes of TMA frequently include renal impairment, potentially necessitating sustained treatment modalities like dialysis or renal transplantation. These conditions exert a profound influence on patient well-being, quality of life, and longevity. Post-remission, it is generally recommended that patients engage in ongoing clinical surveillance. TMA survivors may necessitate extended pharmacological management to address residual dysfunctions and avert recurrence. Notably, enduring sequelae can arise from the neurological manifestations of TMA, necessitating support and consultation with neurologists or rehabilitation experts as part of a comprehensive management plan.

Additionally, given the acute severity of TMA, enduring emotional strain or significant psycho-emotional disturbances, including anxiety spectrum disorders and depressive conditions, may afflict patients. The engagement of psychological or psychiatric services constitutes a critical facet of the recovery and rehabilitation process for these individuals.

## 12. Conclusions

The phenomenon of TMA in oncological patients represents a complex and insufficiently elucidated challenge. Moreover, the incidence of DITMA is anticipated to amplify in conjunction with the diversification of antineoplastic treatments, which includes the utilization of polypharmacy approaches. The pathogenetic mechanisms and therapeutic strategies associated with chemotherapy-induced TMA necessitate additional investigation. The management of such conditions presents considerable difficulties in the absence of established diagnostic and therapeutic protocols. Currently, there is an absence of substantive evidence to advocate for any specific treatments beyond the immediate cessation of the offending pharmacological agent. However, complement inhibition could potentially offer therapeutic advantages in cases that are severe or resistant to standard treatments. The deployment of non-invasive diagnostic instruments, such as biomarkers indicative of endothelial injury, merits consideration for their potential to facilitate precocious diagnosis. Engagement in functional and genetic complement analyses, alongside the exploration of divergent therapeutic interventions within the framework of randomized clinical trials, may augment the evaluation and tailoring of management approaches, thereby enhancing patient prognoses.

Hence, TMA associated with neoplasms, chemotherapeutic agents, or hematopoietic stem cell transplantation represents a critical and potentially fatal condition. The promotion of heightened vigilance, coupled with the early detection and intervention of these disorders, has the potential to curtail associated morbidity and mortality significantly. Novelties in therapeutic options for TMA, when utilized in concert with chemotherapy and transplantation, have yielded notable advancements in patient survival.

## Figures and Tables

**Figure 1 ijms-25-09055-f001:**
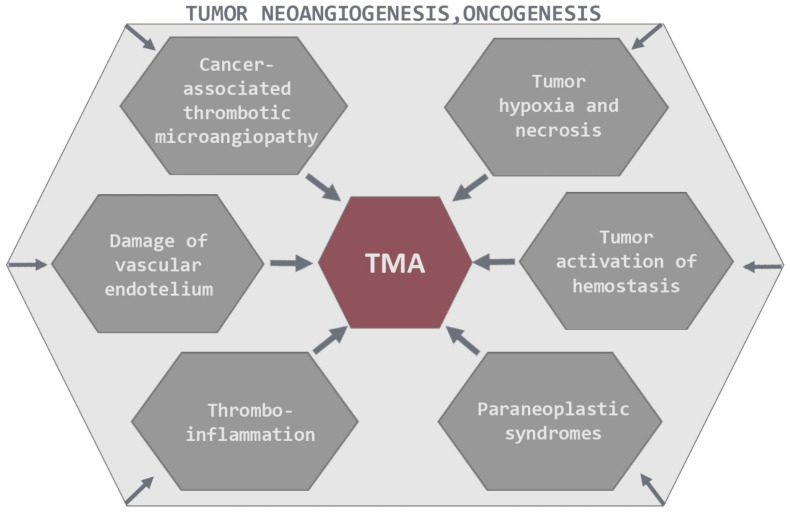
Pathogenesis of TMA in cancer patients.

**Figure 2 ijms-25-09055-f002:**
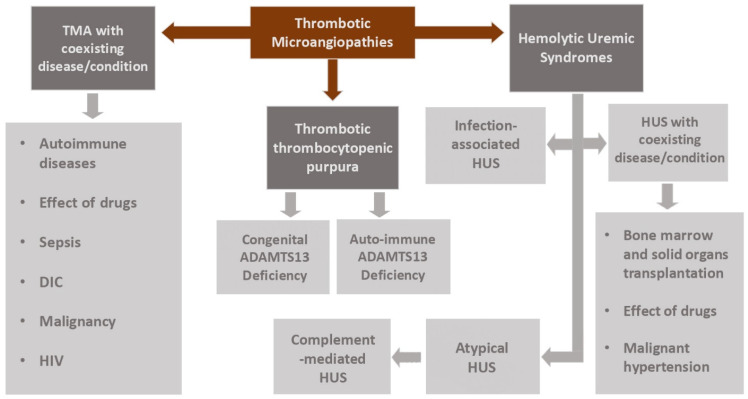
Differential diagnosis of TMA.

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
