# Peer review of "The Phenomenon of Thrombotic Microangiopathy in Cancer Patients"

_ijms, 2024, doi:10.3390/ijms25169055_

Round 1

Reviewer 1 Report

Comments and Suggestions for Authors

Thank you for the invitation to review this work. The manuscript reviews thrombotic microangiopathy (TMA) in cancer patients, covering historical aspects, pathophysiology, clinical presentations, and treatment strategies.

While this is an interesting and informative read, some sections could benefit from further clarity and conciseness. Below are few suggestions for authors to consider;

1. Abstract (Lines 24-43, Page 1): The abstract could be slightly restructured to improve flow. Consider breaking down the abstract into clear sections: Background, Methods, Results, and Conclusion.

2. Introduction (Lines 46-77, Page 2): The introduction provides a good overview of TMA but is quite dense. Suggest simplifying complex sentences for better readability. For example, the sentence in lines 47-49 could be split into two sentences for clarity. 

3. Lines 78-137, Pages 2-3: This section is informative but somewhat lengthy. Suggest condensing the historical information to focus on the most critical developments, ensuring relevance to the current understanding of TMA.

4. Lines 138-186, Pages 4-5): Some information is repetitive. Suggest eliminating redundancy. For example, lines 147-151 repeat information from lines 138-144. Consider merging these points for brevity.

5.  Lines 187-238, Pages 5-6: This section covers complex mechanisms effectively but could benefit from more subheadings to guide the reader. Suggest adding subheadings such as "Role of ADAMTS13" and "Mechanisms in Cancer Patients" to improve readability.

6. Lines 239-339, Pages 7-9): This section could be more detailed. Suggest expanding on the specifics of treatment options, including recent advances in therapy and ongoing research.

7. References: The references are appropriate and relevant, though some recent studies may be missing. Suggest updating the reference list to include the most recent studies in TMA, particularly those published in the last two years.

Comments on the Quality of English Language

Moderate editing is required.

Author Response

Response 1: Abstract (Lines 24-43, Page 1): The abstract could be slightly restructured to improve flow. Consider breaking down the abstract into clear sections: Background, Methods, Results, and Conclusion

Comments 1: Since the presented article is a literature review, the abstract was presented in a standard form for this type of publication

Response 2: Introduction (Lines 46-77, Page 2): The introduction provides a good overview of TMA but is quite dense. Suggest simplifying complex sentences for better readability. For example, the sentence in lines 47-49 could be split into two sentences for clarity

Comments 2: Necessary corrections have been made

Response 3: Lines 78-137, Pages 2-3: This section is informative but somewhat lengthy. Suggest condensing the historical information to focus on the most critical developments, ensuring relevance to the current understanding of TMA.

Comments 3: Necessary corrections have been made

Response 4: Lines 138-186, Pages 4-5): Some information is repetitive. Suggest eliminating redundancy. For example, lines 147-151 repeat information from lines 138-144. Consider merging these points for brevity.

Comments 4: Necessary corrections have been made

Response 5:  Lines 187-238, Pages 5-6: This section covers complex mechanisms effectively but could benefit from more subheadings to guide the reader. Suggest adding subheadings such as "Role of ADAMTS13" and "Mechanisms in Cancer Patients" to improve readability.

Comments 5: Necessary corrections have been made

Response 6: Lines 239-339, Pages 7-9): This section could be more detailed. Suggest expanding on the specifics of treatment options, including recent advances in therapy and ongoing research.

Comments 6: The section on approaches to the treatment of TMA has been significantly revised and supplemented, mainly due to literary sources of the last 2 years

Response 7: References: The references are appropriate and relevant, though some recent studies may be missing. Suggest updating the reference list to include the most recent studies in TMA, particularly those published in the last two years.

Comments 7: Necessary corrections have been made

Reviewer 2 Report

Comments and Suggestions for Authors

Thank you for the opportunity to review your paper.

The study is a review of the literature regarding the TMA.

are any other situations except cancer and Covid -19 with risk of TMA?

how did you  evaluate the improvement of quality of life?

are there any described guidelines for diagnose and treat TMA? 

Author Response

Response 1: The study is a review of the literature regarding the TMA. Are any other situations except cancer and Covid -19 with risk of TMA?

Comments 1: The article is devoted to the features of TMA in cancer patients, but it also highlights the role of pregnancy, septic conditions and COVID-19 as triggers of TMA

Response 2: How did you evaluate the improvement of quality of life?

Comments 2: We did not assess patients' quality of life, this was removed from our conlusions

Response 3: Are there any described guidelines for diagnose and treat TMA? 

Comments 3: The section on approaches to the treatment of TMA has been significantly revised and supplemented, mainly due to literary sources of the last 2 years

Round 2

Reviewer 1 Report

Comments and Suggestions for Authors

No further comments. The revised verison may be accepted.

Comments on the Quality of English Language

Ok

Author Response

Thank you very much for the reply!